# Non-Cholesterol Sterols in Breast Milk and Risk of Allergic Outcomes in the First Two Years of Life

**DOI:** 10.3390/nu14040766

**Published:** 2022-02-11

**Authors:** Lieve van Brakel, Carel Thijs, Ronald P. Mensink, Dieter Lütjohann, Jogchum Plat

**Affiliations:** 1Department of Nutrition and Movement Sciences, NUTRIM School of Translational Research in Metabolism, Maastricht University, 6200 MD Maastricht, The Netherlands; lieve.vanbrakel@maastrichtuniversity.nl (L.v.B.); r.mensink@maastrichtuniversity.nl (R.P.M.); 2Department of Epidemiology, CaPHRI (Care and Public Health Research Institute), Maastricht University, 6200 MD Maastricht, The Netherlands; c.thijs@maastrichtuniversity.nl; 3Institute of Clinical Chemistry and Clinical Pharmacology, University Hospital Bonn, 53127 Bonn, Germany; dieter.luetjohann@ukbonn.de

**Keywords:** non-cholesterol sterols, eczema, wheeze, allergic sensitization, breast milk, immune system, infant health

## Abstract

This study aimed to explore associations between non-cholesterol sterol concentrations in breast milk and allergic outcomes in children aged two. Data from the KOALA Birth Cohort Study, the Netherlands, were used. Non-cholesterol sterols were analyzed by gas–liquid chromatography–mass spectrometry in breast milk sampled one-month postpartum (*N* = 311). Sterols were selected for each allergic outcome, i.e., eczema, wheeze, and allergic sensitization, prior to analyses. Associations between the selected sterols with allergic outcomes were analyzed using multiple logistic regression to calculate odds ratios (ORs). The odds of eczema in the first two years of life were lower with higher concentrations of cholestanol (OR (95%CI): 0.98 (0.95; 1.00), *p* = 0.04), lanosterol (0.97 (0.95; 1.00), *p* = 0.02), lathosterol (0.93 (0.87; 0.99), *p* = 0.02), and stigmasterol (0.51 (0.29; 0.91), *p* = 0.02) in breast milk sampled one-month postpartum. None of the sterols were associated with wheeze in the first two years of life. The odds of allergic sensitization at age two were lower with higher concentrations of campesterol in breast milk (OR (95%CI): 0.81 (0.70; 0.95), *p* = 0.01). In conclusion, our data suggest that exposure to higher non-cholesterol sterol concentrations in breast milk may indeed be associated with the prevention of allergic outcomes in the first two years of life.

## 1. Introduction

Breastfeeding is the preferred nutrition for newborns and infants [1]. The World Health Organization therefore recommends exclusive breastfeeding for the first six months of life and to combine breastfeeding with complementary foods for children aged from six months to two years and beyond [2]. Breastfeeding has several health benefits for infants. For example, breastfeeding has been associated with a decreased risk of child mortality in the first two years of life [3]. In addition, probiotic bacteria in breast milk play an essential role in developing the gut microbiota in early life by seeding the infant gut [4]. Breastfeeding even has health benefits tracking into adulthood. Breastfeeding has been associated with a lower risk of several non-communicable diseases in adults, such as cardiovascular diseases [5,6], obesity [7,8], and type 2 diabetes [7,9]. Moreover, in recent years, there has been increasing interest in the potential role of breastfeeding for the prevention of allergic outcomes in newborns and infants [10,11,12,13,14,15].

Although results are inconclusive [10], breastfeeding has been associated with a reduced risk of developing asthma [11,12,13], eczema [13,14], and allergic diseases [15]. It is therefore important to identify compounds within breast milk that could be responsible for the supposed reduced risk of developing allergic diseases. However, identification of these compounds is difficult, since breast milk composition is highly variable, especially during the first month of breastfeeding [16]. Variability in composition is highest in the milk produced during the first three weeks postpartum: colostrum (produced in first 4–7 days) and transitional milk (produced approximately from day 7 to 21 postpartum) [17]. This variability may reflect the infant’s needs, e.g., for infant growth [18]. The composition of mature milk (produced from approximately day 21 postpartum onwards) is less variable and contains approximately 3–5% (*w*/*w*) fat, 6.9–7.2% carbohydrates, 0.8–0.9% protein, and 0.2% mineral constituents [17,19]. Lipids in breast milk are the most important energy source for infants [16]. The lipid fraction of breast milk mainly consists of triacylglycerol and for approximately 0.5% of cholesterol [20]. It also contains plant sterols, which surprisingly do not reflect the circulating plasma plant sterol concentrations of the mother [21]. In addition, mRNA expression for sterol transporters ABCG5/G8 was previously observed in bovine mammary glands [22]. When this is also the case in human mammary glands, it could explain the presence of the specific plant sterol concentrations in breast milk [21]. Altogether, these findings suggest a regulated transport process of plant sterols into breast milk. 

Today, plant sterols are mainly recognized for their LDL-cholesterol-lowering effects [23]. However, in a paper published by our group, Plat and colleagues have suggested that plant sterols in breast milk may have a perinatal role, e.g., in growth and development of the child [21]. This rationale was, among others, based on that plant sterols have been shown to interact with immune cells [24]. In more detail, plant sterols and stanols (saturated derivatives of plant sterols) may affect T-helper cell behavior, potentially by interacting with regulatory T cells (Tregs) [25,26,27]. This effect of plant sterols and stanols could be relevant in conditions characterized by a disbalance between T-helper cell subsets Th1 and Th2. For instance, a disbalance in T-helper cell activity towards the Th2 profile has been related to increased immunoglobulin E (IgE) concentrations and allergic diseases, such as allergic asthma [28]. Brüll and colleagues studied the effects of plant stanols on immune cells of allergic asthma patients. Based on their in vitro and in vivo observations, they suggested that plant stanols stimulated Treg and Th1 cell activity, while inhibiting Th2 cell activity [25,26]. Furthermore, plant sterols share a structural similarity with cholesterol precursors, which are intermediates in the endogenous cholesterol synthesis pathways [21]. Previous research has shown that some of these compounds can also interact with immune cells, thereby affecting immune responses. For example, desmosterol was found to inhibit inflammatory cascades within macrophages [29]. Moreover, mevalonate, which is another intermediate in the cholesterol synthesis pathway, was found to be crucial to induce trained immunity [30]. As with plant sterols, cholesterol precursors are also present in breast milk [21]. Together, these sterol compounds can be grouped as non-cholesterol sterols. However, it is important to consider these non-cholesterol sterols in breast milk as nutrients and not as markers for intestinal cholesterol absorption and endogenous synthesis for which their serum concentrations have been validated [31].

It is unknown whether the suggested effects of non-cholesterol sterols on immune cell behavior translate in to a benefit for children when exposed to these compounds in early life. This raises the question whether the amount of non-cholesterol sterols in breast milk could influence immune maturation, alter T-helper cell behavior and immune responses in early life, and thereby prevent allergic diseases. Therefore, this study aimed to determine the association between non-cholesterol sterol concentrations in breast milk and allergic outcomes in breastfed children in the first two years of life.

## 2. Materials and Methods

### 2.1. Study Population

The cohort used in this study is part of the “Kind, Ouders en gezondheid: Aandacht voor Leefstijl en Aanleg” (KOALA) Birth Cohort Study, the Netherlands, which has been described in detail elsewhere [32]. Briefly, recruitment of pregnant women started in October 2000. Participants with a conventional (*N* = 2343) or an ‘alternative’ lifestyle (*N* = 491) with regard to, e.g., child rearing practices or diet (organic or vegetarian) were recruited and enrolled between the 14th and 18th week of gestation. Participants were followed during gestation and up to several years postpartum and completed relevant questionnaires during follow-up. From January 2002, we started collecting biosamples, such as maternal blood at 36 weeks of pregnancy. In this subcohort of the KOALA study (KOALA-SUB), other samples were also obtained, including a breast milk sample from the mother one-month postpartum and a venous blood sample from the child at age two. For this study, we used these two samples and the data collected from questionnaires until the age of two. The KOALA study was approved by the Medical Ethical Committee of Maastricht University Medical Center, Maastricht, the Netherlands (MEC 01-139 and 00-182) and the Central Committee on Research Involving Human Subjects, The Hague, the Netherlands (CCMO P01.1265L). Inclusion criteria for the present study were participation in KOALA-SUB and an available one-month postpartum breast milk sample. Exclusion criteria were prematurity (<37 weeks gestation) and diseases or disorders such as cystic fibrosis, Down’s syndrome, and arthritis. The current study included *N* = 311 children (*N* = 141 mothers with conventional lifestyle, *N* = 166 mothers with alternative lifestyle, and *N* = 4 mothers have missing data on lifestyle).

### 2.2. Study Outcomes

Allergic outcomes of interest were eczema and wheeze during the first two years of life, and allergic sensitization at age two. At that age, the immune system has had the opportunity to mature while being exposed to different concentrations of non-cholesterol sterols in breast milk [33]. The International Study of Asthma and Allergies in Childhood Questionnaire (ISAAC) was used to determine the presence of eczema and wheeze at 3, 7, 12 and 24 months postpartum, as described previously [34]. In short, if parents ever reported symptoms of eczema (itchy rash that was coming and going) or wheeze (wheezing or whistling in the chest) in this questionnaire, the child was defined as a case of eczema or wheeze, respectively. Children who only had diaper rash, rash around the eyes, or scalp scaling were not considered to have eczema. Allergic sensitization against hen’s egg, cow’s milk, peanut, birch, grass pollen, cat, dog, or house dust mite was determined in a venous blood sample at age two. Allergic sensitization was defined as having specific serum IgE levels > 0.30 IU/mL against one of the allergens tested. As described earlier, IgE was measured with a detection limit of 0.10 IU/mL [35,36]. 

### 2.3. Breast Milk Sampling and Analysis of Non-Cholesterol Sterols

Methods for breast milk sampling and storage have been described elsewhere [37]. Briefly, breast milk was collected in the morning in sterile tubes (Greiner Bio-One, Kremsmuenster, Austria). A sample was collected from the contra-lateral breast since the last feeding, before breastfeeding the child. The milk samples were kept in the refrigerator (4 °C) and picked up by a researcher on the same day. During transport, the milk samples were stored in a cooler (Coleman Company, Inc., Breda, the Netherlands) on packed ice (4 °C) until processing on the same day at the Biobank Maastricht. After measuring the volume of the sample, it was mixed (gently shaking by hand) and five Eppendorf tubes (2 mL) were filled with whole milk for storage. Two Eppendorf tubes were filled for creamatocrit measurement. The remaining sample was centrifuged (400× *g*, 12 min, no brake, 4 °C) to separate the lipid and aqueous fraction. The lipid layer was trimmed off with a pipette and released in plastic storage vials (Sarstedt, Nümbrecht, Germany) and stored at −80 °C in the Biobank Maastricht until further processing. Creamatocrit was determined as described previously [38,39]. In short, milk samples were centrifuged for 15 min at 12,000× *g*. The length of the total milk column and of the cream layer were measured directly after centrifuging. Creamatocrit was determined by calculating which percentage of the total length of the milk sample consisted of cream.

For cholesterol and non-cholesterol sterol analysis, the frozen breast milk samples were transported on dry ice and delivered on the same day at the Institute of Clinical Chemistry and Clinical Pharmacology of the University Hospital Bonn, Germany. Concentrations of cholesterol were measured by gas-liquid chromatography flame ionization detection (GC-FID) on an HP6890 Series GC-System (Agilent Technologies, Waldbronn, Germany), using 5α –cholestane (Serva Electrophoresis GmbH, Heidelberg, Germany) as internal standard. Cholesterol and non-cholesterol standards were applied by Sigma-Aldrich Chemie GmbH, München, Germany. Plant sterol (sitosterol, campesterol, stigmasterol, brassicasterol), cholestanol and cholesterol precursor (lanosterol, lathosterol and desmosterol) concentrations in breast milk were analyzed by gas–liquid chromatography–mass spectrometry-selected ion monitoring (GC-MS-SIM) (Agilent Technologies 6890 Network GC coupled with an Agilent Technologies 5975B inert MSD, Agilent Technologies, Waldbronn, Germany), using epicoprostanol (sigma-Aldrich Chemie GmbH, München, Germany) as internal standard [40]. Sterol concentrations were corrected for the lipid levels (creamatocrit) of the breast milk sample by dividing the sterol concentrations by creamatocrit expressed as fraction.

### 2.4. Statistical Analysis

Two types of exploratory analyses were first conducted to determine which sterols could be associated with allergic outcomes. First, factor analysis was performed to determine correlations between sterol concentrations in breast milk. A varimax rotation was used to maximize between-subject variance and the minimal eigenvalue was set to 1. The obtained factors were then used in multiple logistic regression analysis to determine factors associated with the allergic outcomes, and to calculate odds ratios (ORs) and their corresponding 95% confidence intervals (95%CIs). Second, independent-sample *t*-tests were performed to determine differences in sterol concentrations between cases and controls for each allergic outcome. Based on these two exploratory analyses, sterols were selected for further analysis with *p* < 0.10 as selection threshold. Sterols were selected when they: (1) were present in factors that were associated with an allergic outcome in the multiple logistic regression analyses (trend [*p* < 0.10] or significant association [*p* < 0.05]), or (2) differed in concentration between cases and controls in the independent-sample *t*-tests (trend [*p* < 0.10] or significant association [*p* < 0.05]). Next, multiple logistic regression models were made for each selected sterol to determine which sterols were significantly associated with allergic outcomes (*p* < 0.05). ORs and corresponding 95%CIs were calculated. A priori, confounders to be used in these regression models were determined by drawing and analyzing causal diagrams (DAGs). Confounders that were tested in the models included study group, smoking, season of milk sampling, gestational age, prepregnancy BMI, maternal age, atopy of parents, maternal education, gender of child, gravidity, duration of breastfeeding, and birthweight. If confounders changed the regression coefficient β1 by at least 10%, they were added to regression models. Finally, Spearman correlations were used to explore relations between non-cholesterol sterol and cholesterol concentrations in breast milk.

A result was considered significantly different when *p* < 0.05. All analyses were conducted using IBM SPSS Statistics for Windows, Version 25.0 (Armonk, NY, USA).

## 3. Results

### 3.1. Baseline Characteristics and Flow Chart

The selection of participants from the KOALA study is shown in Figure 1. Of the total cohort (*N* = 2834), the women with an available breast milk sample were selected (*N* = 315). The 311 women who fulfilled the criteria for the current study were selected.

Characteristics of the study population are shown in Table 1. The mean (SD) maternal age and median BMI (IQR) of the 311 mothers at the start of pregnancy were 32.4 (3.9) years and 22.4 (20.6–24.5) kg/m^2^, respectively. In total, 91 children developed eczema and 79 children developed wheeze in the first two years of life, and 49 children were allergically sensitized against common allergens at age two. The baseline characteristics were comparable between the women with either a conventional or alternative lifestyle. Only sitosterol was higher in the alternative lifestyle group (Table 1).

Desmosterol was the non-cholesterol sterol with the highest concentration in breast milk (median (IQR): 52.2 (37.4–70.3) µmol/L), which was 25 to 1000 fold higher as compared to the other sterols. Stigmasterol was the sterol with the lowest concentration (0.05 (0.04–0.06) µmol/L). Overall, non-cholesterol concentrations in breast milk were similar in the women with a conventional lifestyle and an alternative lifestyle. 

### 3.2. Selection Process of Sterols

#### 3.2.1. Exploratory Factor Analysis

To explore which of the eight non-cholesterol sterols that were analyzed in breast milk were intercorrelated, exploratory factor analysis was performed. Two factors were found based on the sterol concentration in breast milk corrected for creamatocrit (Table 2). The two factors separated brassicasterol, stigmasterol, campesterol, and lathosterol (factor 1); and lanosterol and desmosterol (factor 2). Cholestanol and sitosterol loaded on both factors, although to a higher extent on factor 1 than on factor 2. 

#### 3.2.2. Multiple Logistic Regression Using the Factors

Multiple logistic regression was performed to explore relations between factors 1 and 2 with the allergic outcomes of interest (i.e., eczema and wheeze in the first two years of life, and allergic sensitization at age two) (Table 3). None of the factors were significantly associated with eczema, wheeze, or allergic sensitization. However, trends were observed for associations between factor 1 and eczema (OR (95%CI): 0.69 (0.46; 1.03), *p* = 0.07), factor 2 and eczema (0.69 (0.46; 1.04), *p* = 0.08), and factor 1 and allergic sensitization (0.52 (0.26; 1.07), *p* = 0.07).

#### 3.2.3. Independent-Sample *t*-Tests

Independent-sample *t*-tests were performed to explore which of the individual sterols differed between cases and controls for each allergic outcome (Table 4). Lathosterol (*p* = 0.06) and stigmasterol (*p* = 0.08) concentrations in breast milk tended to be lower in eczema cases compared to controls. For wheeze, all sterol concentrations were similar in cases and controls. For allergic sensitization, campesterol concentrations in breast milk were significantly lower in cases compared to controls (*p* = 0.03).

### 3.3. Multiple Logistic Regression Using Selected Sterols

#### 3.3.1. Eczema

Based on the multiple logistic regression analysis using the obtained factors (Table 3), all sterols included in factors 1 and 2 were selected for eczema. In addition, based on the independent-sample *t*-tests (Table 4), lathosterol and stigmasterol were selected for eczema. Thus, all eight sterols were included in the final multiple logistic regression analyses. Separate models were made for each individual sterol (Table 5). The odds of eczema in the first two years of life were significantly lower with higher concentrations of cholestanol (OR (95%CI): 0.98 (0.95; 1.00), *p* = 0.04), lanosterol (0.97 (0.95; 1.00), *p* = 0.02), lathosterol (0.93 (0.87; 0.99), *p* = 0.02), and stigmasterol (0.51 (0.29; 0.91), *p* = 0.02) in breast milk one-month postpartum. The other sterols did not affect the odds of eczema during the first two years of life.

#### 3.3.2. Wheeze

None of the factors from exploratory factor analysis were associated with wheeze, nor were there differences in sterol concentrations in breast milk between cases and controls. Therefore, none of the sterols were evaluated in further analysis for wheeze.

#### 3.3.3. Allergic Sensitization

Based on the multiple logistic regression analysis using the obtained factors (Table 3), the sterols included in factor 1 were selected for allergic sensitization. In addition, based on the independent-sample *t*-tests (Table 4), campesterol was selected for allergic sensitization. Thus, brassicasterol, campesterol, cholestanol, lathosterol, sitosterol, and stigmasterol were included in the final multiple logistic regression analyses. Separate models were made for each individual sterol (Table 6). The odds of allergic sensitization at age 2 were significantly lower with a higher concentration of campesterol in breast milk one-month postpartum (OR (95%CI): 0.81 (0.70; 0.95), *p* = 0.01). The other sterols did not affect the odds of allergic sensitization at age 2.

### 3.4. Cholesterol and Allergic Outcomes

Non-cholesterol sterol concentrations were significantly correlated to cholesterol concentrations (corrected for creamatocrit) in breast milk, except for lanosterol and lathosterol concentrations (Table 7). Therefore, relationships between cholesterol concentrations (corrected for creamatocrit) in breast milk and allergic outcomes were also considered. However, the odds of having eczema or wheeze in the first two years of life were not lower with higher cholesterol concentrations, nor were the odds for allergic sensitization at age two (data not shown). Hence, non-cholesterol sterols did not act as a marker for cholesterol. The reported associations can instead be attributed specifically to the non-cholesterol sterols.

## 4. Discussion

The aim of this study was to determine the association between non-cholesterol sterols in breast milk and allergic outcomes in breastfed children in the first two years of life. We found that the odds of eczema during the first two years of life were significantly lower with higher concentrations of cholestanol, lanosterol, lathosterol, and stigmasterol in breast milk one-month postpartum. We also showed that the odds of allergic sensitization at age 2 were significantly lower with a higher concentration of campesterol in breast milk. None of the sterols were associated with wheeze during the first two years of life. Study groups (women with a conventional or ‘alternative’ lifestyle with regard to, e.g., child rearing practices) were not further compared, since study group did not seem to influence the reported associations. A priori, we hypothesized that exposure of the immune system to non-cholesterol sterols through breastfeeding early in life influences the maturation of the immune system and thereby prevents allergic outcomes later in life. Our results presented here support this hypothesis and are in line with previous suggestions that non-cholesterol sterols may play a role in infant health [21]. 

Non-cholesterol sterols in serum are known for their relationship with cholesterol metabolism [21,23,24]. This group of sterols can be divided into two subgroups, i.e., some are diet derived and considered as markers for intestinal cholesterol absorption (brassicasterol, campesterol, cholestanol, sitosterol, stigmasterol), while others are endogenously synthesized and markers for cholesterol synthesis (desmosterol, lanosterol, lathosterol). However, these two subgroups were not identified when exploratory factor analysis was performed using concentrations of these non-cholesterol sterols in breast milk (with or without correction for creamatocrit). This finding is in line with our hypothesis that sterols provided by breast milk should be considered as nutrients (and not as markers for intestinal cholesterol absorption and endogenous synthesis), which may have specific effects in the body in early life, e.g., involvement in the maturation of the immune system. In addition, studies in adults have also reported effects of non-cholesterol sterols on the immune system. Brüll and colleagues [25] used antibody production to a hepatitis A vaccine as a measure for immune function in adult asthma patients that received either plant stanols or placebo. They reported that daily intake of 4 g of plant stanols increased antibody production by 22% compared to placebo [25]. In addition, changes in serum plant stanol concentrations were positively correlated to the Th1/Th2 cytokine balance towards more Th1 activity [25]. These results, together with our current findings, indicate that consuming plant sterols and stanols may not only affect cholesterol metabolism, but may also be related to developing and sustaining immune function throughout life.

Results of studies evaluating the effect of breastfeeding on allergic outcomes in children are inconclusive, and information on the maternal diet during breastfeeding is often missing [41]. Therefore, a clear recommendation for future studies is to include data about maternal diet composition during pregnancy and breastfeeding, and/or breast milk composition. Unfortunately, no studies have related non-cholesterol sterols in breast milk and infant feeding with allergic outcomes in children, which makes it difficult to compare our study results. However, results can be compared with studies evaluating the effect of children’s intake of diets or foods rich in these non-cholesterol sterols on allergic outcomes. Our results are in accordance with a review, which stated that plant-based diets and diets similar to the Mediterranean diet, which are generally rich in plant sterols, could reduce inflammation and asthma symptoms in children [42]. Another study reported a negative association between fruit and vegetable intake and allergic symptoms in children [43]. Moreover, the results of our study could be compared to studies evaluating the effects of non-cholesterol sterols on immune-related outcomes in other parts of the body, such as the gut. Van Gorp and colleagues found that intra amniotic administration of β-sitosterol and campesterol prevented gut inflammation in fetal lambs that were intra-amniotically infected with *Ureaplasma parvum* [44]. Plasma IL-6, influx of mucosal myeloperoxidase-positive cells, and intestinal damage were all lowered by the intra amniotic administration of plant sterols [44]. In addition, de Smet and colleagues showed that an acute intake of plant stanols down-regulated genes regulating T-cell functioning in the jejunum of healthy volunteers [45]. These two studies also indicate that non-cholesterol sterols are able to influence immune cell behavior, although the exact mechanisms remain unclear.

Considering our results and the studies described above, it is tempting to suggest that higher intakes of non-cholesterol sterols via breast milk would result in better health outcomes. Infant formulas sometimes also contain high concentrations of plant sterols, especially when produced with vegetable oil as fat source [46], while formula feeding has not been associated to better health outcomes compared to breastfeeding [47,48,49]. Claumarchirant and colleagues [46] reported that total plant sterol concentrations (sum of brassicasterol, campesterol, β-sitosterol, stigmasterol, and sitostanol) in various infant formulas ranged between 3.1–5.0 mg/100 mL (78.1–132.3 µmol/L). These concentrations are higher than those reported in our study, where the median total plant sterol concentration in breast milk (sum of brassicasterol, campesterol, sitosterol, and stigmasterol) was 1.3 µmol/L. However, desmosterol (0.2–0.4 mg/100 mL (6.2–11.1 µmol/L)) and cholesterol (1.6–5.1 mg/100 mL (0.04–0.1 mmol/L)) concentrations in infant formulas [46] were lower as compared to the concentrations we found in breast milk (desmosterol: 52.2 µmol/L; cholesterol 0.35 mmol/L). Hence, not only absolute concentrations, but also ratios between individual sterols differ between breast milk and infant formula. In more detail, the ratio between cholesterol and sitosterol concentrations in infant formula ranges from approximately 0.6 to 1.8 [46], whereas in our study the median cholesterol concentration was 500 times higher than the sitosterol concentration (350 µmol/L and 0.70 µmol/L, respectively). The lower concentration of cholesterol in infant formulas as compared to breast milk induces higher endogenous cholesterol synthesis in formula fed infants, whereas breastfed infants have a higher intestinal cholesterol absorption [50,51]. To the best of our knowledge, the bioavailability of non-cholesterol in breast milk and infant formula has not been studied. Therefore, it is currently not completely understood how serum non-cholesterol concentrations in children are affected by either breastfeeding or formula feeding. 

Additionally, it is currently unknown whether the differences in sterol concentrations between breast milk and infant formula are associated with the difference in the immune responses of breastfed versus formula fed children. However, it should be kept in mind that breast milk and infant formula differ in many more aspects that could affect immune system development than solely non-cholesterol sterol and cholesterol concentrations. For example, breast milk contains human immune factors, which help forming the neonatal immune system [52]. Ultimately, the relation between early infant feeding and allergic outcomes is not fully understood. Future studies should further evaluate: (1) whether the reported effects of non-cholesterol sterols on allergic outcomes in our study can be attributed to their concentration in breast milk, (2) whether the ratio between the different sterols could also play a role, and (3) whether other components in breast milk are potentially involved in this association.

Although the associations between sterols in breast milk, eczema, and allergic sensitization were statistically significant, none of the sterols were significantly associated with wheeze in the first two years of life. There are several wheezing phenotypes, based on the age at which wheezing first occurs [53]. For example, phenotypes such as transient early wheeze and prolonged early wheeze are characterized by wheezing only in the first years of life, while wheezing disappears as the child gets older. Other phenotypes such as intermediate onset wheeze or late onset wheeze are characterized by wheezing occurring at a later age (18–42 months old). The phenotypes characterized by later onset wheeze are strongest associated with allergic outcomes later in life [53]. For this study, it means some of the children could have suffered from the phenotypes characterized by early onset wheeze, which could also be caused by viral infections [54]. It would be interesting to evaluate the association between non-cholesterol sterols in breast milk and wheezing or even asthma at a later age, and to take wheezing phenotypes into account. Unfortunately, the number of late onset wheeze and asthma cases in the subgroup with breast milk samples was insufficient to allow proper statistics. The associations between non-cholesterol sterols, wheeze, and asthma should therefore be evaluated in a larger study. 

Another limitation of this study was the extensive selection of sterols and that multiple allergic outcomes were tested. The exploratory nature of this study may have increased the chance of type I errors. Therefore, data should be interpreted with care and additional studies are needed to confirm or refute our findings. In addition, future studies should consider whether there are optimal sterol concentrations in breast milk, whether ratios between different sterols in breast milk play a role in the prevention of allergic outcomes, and how breast milk composition fits into this association. 

## 5. Conclusions

In conclusion, our data suggest that exposure to higher non-cholesterol sterol concentrations (corrected for creamatocrit) in breast milk may indeed contribute to the prevention of allergic outcomes, such as eczema and allergic sensitization at the age of two. Evidence regarding the elaborate role of sterols in human health rapidly grows and should be explored in further detail. Future studies should consider a role for breast milk composition and maternal diet during pregnancy and lactation in the association between breastfeeding and allergic outcomes in children. The effects of sterol intake via breastfeeding versus bottle feeding on allergic disease prevention should also be studied in more detail. 

## Figures and Tables

**Figure 1 nutrients-14-00766-f001:**
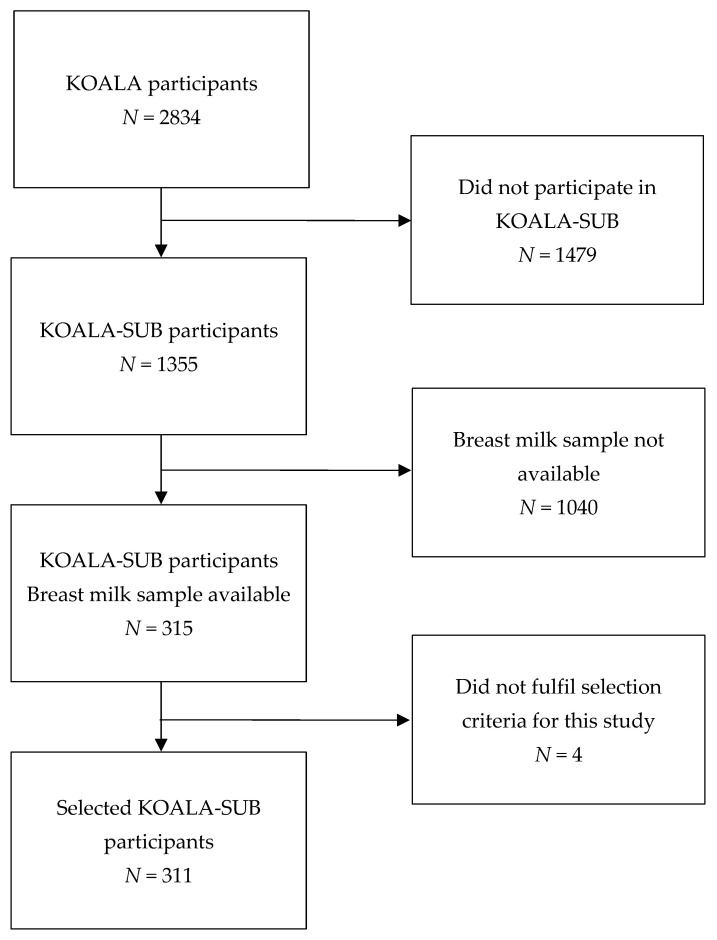
Flow chart of participants included in this study. KOALA = “Kind, Ouders en gezondheid: Aandacht voor Leefstijl en Aanleg”; KOALA-SUB = subcohort of the KOALA study.

**Table 1 nutrients-14-00766-t001:** Baseline characteristics of the KOALA-SUB cohort selected for this study. Data are shown as the mean (SD) or the median (Q1–Q3), unless otherwise indicated.

	Total (*N* = 311)	Conventional Lifestyle ^1^ (*N* = 141)	Alternative Lifestyle ^1^ (*N* = 166)
Maternal age, years (SD)	32.4 (3.9)	31.5 (3.4)	33.1 (4.2)
Prepregnancy BMI, kg/m^2^ (IQR)	22.4 (20.6–24.5)	23.0 (21.5–25.2)	21.7 (20.1–24.0)
Smoking during pregnancy ^2^, N (%)	5 (2%)	4 (3%)	1 (1%)
Atopic history parents ^1^, N (%)			
None	113 (37%)	50 (36%)	62 (38%)
Only father	78 (25%)	37 (26%)	40 (24%)
Only mother	64 (21%)	31 (22%)	32 (20%)
Both	52 (17%)	22 (16%)	30 (18%)
Gender child female, N (%)	161 (52%)	74 (53%)	83 (50%)
Duration breastfeeding, N (%)			
1–3 months	64 (21%)	46 (33%)	17 (10%)
4–6 months	70 (23%)	40 (28%)	29 (18%)
7–9 months	70 (23%)	28 (20%)	41 (25%)
10–12 months	53 (17%)	17 (12%)	36 (22%)
≥13 months	53 (17%)	10 (7%)	43 (26%)
Maternal education, N (%)			
Lower	12 (4%)	7 (5%)	5 (3%)
Middle	96 (31%)	53 (38%)	41 (25%)
Higher vocational	131 (42%)	60 (43%)	69 (42%)
Academic	66 (21%)	19 (14%)	47 (28%)
Other	6 (2%)	2 (1%)	4 (2%)
Season breast milk sampling ^1^, N (%)			
December 2002–February 2003	112 (37%)	64 (45%)	48 (29%)
March–May 2003	120 (39%)	59 (42%)	61 (37%)
June–September 2003	75 (24%)	18 (13%)	57 (34%)
Gravidity, N (%)			
1	123 (40%)	65 (46%)	58 (34.9)
2	110 (35%)	51 (36%)	57 (34.3)
≥3	78 (25%)	25 (18%)	51 (30.7)
Eczema in first two years ^3^, N (%)	91 (30%)	41 (30%)	50 (31.1)
Wheeze in first two years ^4^, N (%)	79 (26%)	37 (27%)	42 (26%)
Allergic sensitization at age 2 ^5^, N (%)	49 (24%)	25 (28%)	24 (22%)
Creamatocrit value ^6^, % (IQR)	7 (5–9%)	7 (5–9%)	7 (5–9%)
Cholesterol concentration breast milk, mmol/L (IQR)	0.35 (0.28–0.42)	0.35 (0.27–0.43)	0.36 (0.29–0.41)
Cholesterol concentration breast milk corrected for creamatocrit ^7^, mmol/Lf (IQR)	4.81 (4.14–5.90)	4.81 (4.17–5.93)	4.83 (5.84–6.87)
Non-cholesterol sterol concentrations breast milk, µmol/L (IQR)			
Brassicasterol	0.23 (0.18–0.27)	0.24 (0.19–0.29)	0.22 (0.17–0.25)
Campesterol ^7^	0.32 (0.20–0.52)	0.37 (0.21–0.60)	0.28 (0.19–0.46)
Cholestanol	1.51 (1.28–1.73)	1.50 (1.26–1.75)	1.52 (1.31–1.73)
Desmosterol	52.2 (37.4–70.3)	51.7 (36.3–68.3)	54.3 (38.1–71.8)
Lanosterol	2.03 (1.43–2.89)	2.01 (1.31–2.85)	2.04 (1.50–2.93)
Lathosterol	0.62 (0.40–0.84)	0.62 (0.38–0.87)	0.62 (0.40–0.84)
Sitosterol	0.70 (0.49–1.41)	0.59 (0.45–0.87)	0.82 (0.55–1.48)
Stigmasterol	0.05 (0.04–0.06)	0.05 (0.04–0.06)	0.05 (0.04–0.06)
Non-cholesterol sterol concentrations breast milk corrected for creamatocrit, µmol/Lf (IQR)			
Brassicasterol ^7^	3.18 (2.44–4.16)	3.32 (2.53–4.25)	2.95 (2.31–4.10)
Campesterol ^8^	4.90 (3.01–7.07)	5.34 (3.50–7.87)	4.27 (2.61–6.13)
Cholestanol ^7^	21.0 (17.2–27.7)	22.5 (17.3–27.9)	20.5 (17.0–27.9)
Desmosterol ^7^	776.5 (592.7–997.9)	716.1(561.6–957.0)	807.2 (633.6–1026.5)
Lanosterol ^7^	29.1 (22.7–37.5)	28.4 (22.8–37.3)	29.7 (22.7–37.5)
Lathosterol ^7^	8.69 (6.25–11.9)	8.80 (6.47–12.2)	8.31 (5.99–11.6)
Sitosterol ^7^	10.3 (7.17–18.7)	8.26 (6.60–12.6)	13.7 (7.88–23.8)
Stigmasterol ^7^	0.68 (0.51–0.93)	0.67 (0.51–0.93)	0.69 (0.51–0.94)

BMI = body mass index; IQR = interquartile range; SD = standard deviation; mmol/Lf = mmol per liter milk fat; ^1^ Missing data *N* = 4; ^2^ Missing data *N* = 1; ^3^ Missing data *N* = 12; ^4^ Missing data *N* = 9; ^5^ Missing data *N* = 110; ^6^ Missing data *N* = 37; ^7^ Missing data *N* = 8; ^8^ Missing data *N* = 45.

**Table 2 nutrients-14-00766-t002:** Factor loadings after varimax rotation. Non-cholesterol sterol concentrations were corrected for creamatocrit.

Non-Cholesterol Sterol (µmol/Lf)	Factor 1	Factor 2
Cholestanol	0.88	0.34
Brassicasterol	0.83	
Stigmasterol	0.78	
Sitosterol	0.65	0.31
Campesterol	0.64	
Lathosterol	0.37	
Lanosterol		0.93
Desmosterol		0.85

µmol/Lf = µmol per liter milk fat; only factor loadings ≥ 0.30 are shown; there were no factor loadings ≤ −0.30.

**Table 3 nutrients-14-00766-t003:** Results of the multiple logistic regression analyses to determine relations between the factors and allergic outcomes.

	Factor 1			Factor 2		
Outcome Variable	OR	95% CI	*p*-Value	OR	95% CI	*p*-Value
Eczema (*N* = 256) ^1^	0.69	0.46; 1.03	0.07 *	0.69	0.46; 1.04	0.08 *
Wheeze (*N* = 259) ^2^	1.04	0.77; 1.40	0.82	0.99	0.69; 1.42	0.95
Allergic sensitization (*N* = 171) ^3^	0.52	0.26; 1.07	0.07 *	1.17	0.79; 1.73	0.43

OR: odds ratio; 95% CI: 95% confidence interval; ^1^: adjusted for season, atopy of parents, maternal education, and duration of breastfeeding; ^2^: adjusted for smoking, season, gestational age, prepregnancy BMI, atopy of parents, maternal education, gender of child, gravidity, and duration of breastfeeding; ^3^: adjusted for study group, smoking, season, prepregnancy BMI, atopy of parents, maternal education, gender of child, gravidity, duration of breastfeeding, and birthweight. * *p* < 0.10.

**Table 4 nutrients-14-00766-t004:** Results of the independent-sample *t*-tests to explore differences between non-cholesterol sterol concentrations in breast milk between cases and controls for eczema (*N* = 80 cases, *N* = 179 controls), wheeze (*N* = 67 cases, *N* = 195 controls), and allergic sensitization (*N* = 41 cases, *N* = 130 controls). Non-cholesterol sterol concentrations were adjusted for creamatocrit.

	Eczema			Wheeze			Allergic Sensitization		
Sterol (µmol/Lf)	Mean Difference	95% CI	*p*-Value	Mean Difference	95% CI	*p*-Value	Mean Difference	95% CI	*p*-Value
Brassicasterol	−0.11	−0.76; 0.55	0.75	0.16	−0.52; 0.84	0.64	−0.25	−1.06; 0.55	0.54
Campesterol	−0.76	−1.85; 0.32	0.17	0.11	−1.03; 1.26	0.84	−1.78	−3.40; −0.16	0.03 **
Cholestanol	−3.19	−7.33; 0.94	0.13	0.3	−4.08; 4.67	0.89	−2.68	−8.04; 2.68	0.33
Desmosterol ^1^	−0.04	−0.14; 0.06	0.44	0.01	−0.11; 0.13	0.83	−0.05	−0.22; 0.12	0.57
Lanosterol	−2.03	−5.55; 1.48	0.26	−0.36	−4.82; 4.10	0.88	0.94	−4.77; 6.65	0.75
Lathosterol	−1.18	−2.39; 0.03	0.06 *	−0.93	−2.23; 0.36	0.16	−0.53	−2.22; 1.16	0.54
Sitosterol	−0.13	−0.28; 0.03	0.11	−0.1	−0.26; 0.07	0.25	−0.13	−0.31; 0.04	0.14
Stigmasterol	−0.01	−0.02; 0.00	0.08 *	0	−0.01; 0.01	0.89	−0.01	−0.02; 0.01	0.36

95% CI: 95% confidence interval; µmol/Lf = micromoles per liter milk fat; ^1^ concentration in mmol/Lf; * *p* < 0.10; ** *p* < 0.05.

**Table 5 nutrients-14-00766-t005:** Results of the multiple logistic regression analyses using the selected sterols for eczema. Separate models were made for each individual sterol. Non-cholesterol sterol concentrations were adjusted for creamatocrit.

Sterol (µmol/Lf)	*N*	OR	95% CI	*p*-Value
Brassicasterol ^1^	264	0.95	0.87; 1.03	0.22
Campesterol ^2^	259	0.95	0.88; 1.03	0.25
Cholestanol ^3^	264	0.98	0.95; 1.00	0.04 *
Desmosterol ^a,1^	264	0.52	0.22; 1.22	0.13
Lanosterol ^4^	267	0.97	0.95; 1.00	0.02 *
Lathosterol ^5^	267	0.93	0.87; 0.99	0.02 *
Sitosterol ^6^	264	0.98	0.95; 1.00	0.09
Stigmasterol ^6^	264	0.51	0.29; 0.91	0.02 *

OR: odds ratio; 95% CI: 95% confidence interval; µmol/Lf = micromoles per liter milk fat; ^a^: unit is mmol/Lf; ^1^: adjusted for season, atopy of parents, maternal education, duration of breastfeeding, and gravidity; ^2^: adjusted for season and maternal education; ^3^: adjusted for atopy of parents and maternal education; ^4^: adjusted for season, maternal education, and duration breastfeeding; ^5^: adjusted for maternal education; ^6^: adjusted for season, atopy of parents, and maternal education; * *p* < 0.05.

**Table 6 nutrients-14-00766-t006:** Results of the multiple logistic regression analysis using the selected sterols for allergic sensitization. Non-cholesterol sterol concentrations were adjusted for creamatocrit.

Sterol (µmol/Lf)	N	OR	95% CI	*p*-Value
Brassicasterol ^1^	176	0.93	0.78; 1.12	0.47
Campesterol ^2^	171	0.81	0.70; 0.95	0.01 *
Cholestanol ^3^	176	0.98	0.95; 1.01	0.26
Lathosterol ^4^	176	0.99	0.91; 1.07	0.77
Sitosterol ^5^	176	0.97	0.93; 1.01	0.13
Stigmasterol ^5^	176	0.77	0.42; 1.40	0.38

OR: odds ratio; 95% CI: 95% confidence interval; µmol/Lf = micromoles per liter milk fat; ^1^: adjusted for study group, season, and duration breastfeeding; ^2^: adjusted for season and duration breastfeeding; ^3^: adjusted for smoking; ^4^: adjusted for smoking, season, and duration breastfeeding; ^5^: adjusted for smoking and season * *p* < 0.05.

**Table 7 nutrients-14-00766-t007:** Spearman correlations between cholesterol and non-cholesterol sterols, *N* = 311.

Sterol (µmol/Lf)	Spearman’s ρ Cholesterol (mmol/Lf)
Brassicasterol	−0.42 **
Campesterol ^1^	0.40 **
Cholestanol	−0.45 **
Desmosterol	−0.13 *
Lanosterol	−0.07
Lathosterol	−0.04
Sitosterol	−0.37 **
Stigmasterol	−0.51 **

µmol/Lf = micromoles per liter milk fat; ^1^ *N* = 303; * *p* < 0.05; ** *p* < 0.001.

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
