# Peer review of "Non-Cholesterol Sterols in Breast Milk and Risk of Allergic Outcomes in the First Two Years of Life"

_nutrients, 2022, doi:10.3390/nu14040766_

Round 1

Reviewer 1 Report

This study by van Brakel et al. addresses an unexplored area of effects of non-cholesterol sterols on developing human immune system. Based on data from the KOALA-SUB subcohort including more than three hundred mothers and their children, the authors tried to describe the link between the content of sterol content in breast milk and immune allergy response in the siblings. The study is well performed; still there are several issues to be resolved:

Were the power analyses for the statistical methods done?

The Table 1 presents also some potentially confounding variables, such as duration of breastfeeding, breast milk sampling season, gravidity. The results in Table 5 list OR for noncholesterol sterols, but adjustment is not done for full spectrum of these factors. Furthermore, it appears that some of the GC-MS analyses were not able to cover all the noncholesterol sterols, even the major one, campesterol.

Some columns in Tables 1, 4 are too narrow to present data in one line. Ref 19 is in the correct format?

Reviewer 2 Report

Prevalence of allergic diseases in childhood and its negative correlation to breastfeeding propels an important investigation of breast milk components. Brakel and colleagues studied association of plant sterols and cholesterol precursors present in breast milk with selected allergies, namely eczema, wheeze and allergic sensitization development. Authors presented a simple approach - sterol content of breast milk was analysed and correlated to allergic outcomes of interest. Results are straightforward without any deeper analysis on why specific cholesterol synthesis precursors and some plant sterols influence maturation of the immune system. The manuscript is well written and requires only minor corrections prior to publishing in Nutrients. 

Main data of this study (Table 1) are presented in two subgroups - “conventional lifestyle” and “alternative lifestyle”. The Discussion chapter, however, lacks any comparison of the two subgroups. This chapter is also quite extensive and I suggest condensing some parts (e.g. infant formulas). 

Rows in Table 1 are incorrectly aligned. 

First column in Table 4 should be wider to avoid splitting the sterol names. 

Reviewer 3 Report

The study is adequately designed and performed. Unfortunately, data presentation is flawed by information not related to the study goals. In any case, the text should be shortened considerably by focusing on the real research interest.

Introduction, line 61 f. It is somehow curious not to clearly mention that the rationale (Plat et al) to perform the present study is published by the own group.  

Reviewer 4 Report

The manuscript by Lieve van Brakel et al. is an interesting article, which reveals the possible relationship between risk of allergic outcomes and content of non-cholesterol sterols in human  milk. 

From the methodological point of view the study seems adequately appropriate and suitable to scientific rigor. The introduction was great to write, the results are clearly reported; the statistical analysis seems also valid and the scientific contents are supported by valid References and the narrative is comprehensive and adequately clear/understandable to common readers; figures and tables are sufficiently clear.

Below, I present several concerns that might help improve the manuscript.

  1. Please consider adding the “Flow chart” of participants included in the study. 
  2. 2. In Table 1. The text such as “Cholesterol concentration breast milk, mmol/L (IQR)” should be the entire line
  3. In all Tables; The description and explanation of the tables should appear below the table and not be part of the table title.
  4. Why do the authors decide to consider results with a p value of less than 0.1 as statistically significant (Usually, statistically significant values below 0.05 are taken)? Moreover, “p” should be in italics (i.e. p-value), please correct throughout the manuscript
  5. Line 304, 319; Please add references.

Overall the manuscript is well-written with no fundamental flows.

Author Response

Reviewer’s comments to author: Reviewer 1
We thank the reviewer for the compliment on how the study was performed.

Question 1: Were the power analyses for the statistical methods done?
Answer: We did not do power analyses for the statistical methods. This is the first study to explore the associations between non-cholesterol sterol concentrations in breast milk and allergic outcomes in children. Therefore, it is difficult to estimate an effect size on forehand. To compensate for this, we chose to select sterols for each allergic outcome using the selecting process as described in the method section (Section 3.2 Selection process of sterols).

Question 2: The Table 1 presents also some potentially confounding variables, such as duration of breastfeeding, breast milk sampling season, gravidity. The results in Table 5 list OR for noncholesterol sterols, but adjustment is not done for full spectrum of these factors.
Answer: That is correct. As described in the method section (line 171 – line 177), we drew and analyzed causal diagrams (DAGs) to determine potentially confounding variables. If these variables changed the regression coefficient β1 by at least 10%, they were added to the regression models. This also means that not all of the potential confounders were added to all of the regression models, but we specifically added confounders per model. The specific confounders per model can be found below each Table.

Question 3: Furthermore, it appears that some of the GC-MS analyses were not able to cover all the noncholesterol sterols, even the major one, campesterol.
Answer: Campesterol is indeed one of the major non-cholesterol sterols in the diet and in serum samples. However, for this study we used breast milk samples, and the distributions between the non-cholesterol sterols in breast milk appear different from plasma samples. As can be seen in Table 1, desmosterol concentrations are highest in breast milk samples. Regarding campesterol, unfortunately some of the samples had very low campesterol concentrations, even below the detection limits of the GC-MS method.

Question 4: Some columns in Tables 1, 4 are too narrow to present data in one line.
Answer: The format of the Tables is not how we submitted them. We have readjusted the width of the columns of Tables 1 and 4 to make sure the data are presented on 1 line.

Question 5: Ref 19 is in the correct format?
Answer: Thank you for noticing this, it was not correct. We have updated the format of Ref 19 now.

Reviewer’s comments to author: Reviewer 2
We thank the reviewer for their compliments about the article.

Question 1: Main data of this study (Table 1) are presented in two subgroups - “conventional lifestyle” and “alternative lifestyle”. The Discussion chapter, however, lacks any comparison of the two subgroups.
Answer: We indeed presented baseline characteristics for two subgroups (conventional vs. alternative lifestyle), since these subgroups were part of the recruitment strategies of the KOALA cohort. We did not make a comparison between subgroups in the discussion, because this variable was only added to
the regression model for allergic sensitization and brassicasterol and therefore did not seem to influence the reported associations. We have now added this comment to the discussion (line 323-325).

Question 2: This chapter is also quite extensive and I suggest condensing some parts (e.g. infant formulas).
Answer: We have now condensed parts of the discussion (in the paragraph now consisting of line 396 – 407).

Question 3: Rows in Table 1 are incorrectly aligned. First column in Table 4 should be wider to avoid splitting the sterol names.
Answer: The format of the Tables is not how we submitted them. We have adjusted the width of the columns of Tables 1 and 4 to make sure the data are presented on 1 line.

Reviewer’s comments to author: Reviewer 3
We thank the reviewer for the compliments about the study design and performance.

Question 1: Unfortunately, data presentation is flawed by information not related to the study goals. In any case, the text should be shortened considerably by focusing on the real research interest.
Answer: We have condensed the discussion paragraph (in the paragraph now consisting of line 396 – 407). We chose to keep the result section as it is, since leaving out parts of this section (or moving it to a supplement) would interfere with the structure of our paper.

Question 2: Introduction, line 61 f. It is somehow curious not to clearly mention that the rationale (Plat et al) to perform the present study is published by the own group.
Answer: We have added this comment to the Introduction (line 61).

Reviewer’s comments to author: Reviewer 4
We thank the reviewer for the compliments about the article.

Question 1: Please consider adding the “Flow chart” of participants included in the study.
Answer: Thank you for this suggestion, we now have included a flow chart of participants included in the study (Figure 1).

Question 2: In Table 1. The text such as “Cholesterol concentration breast milk, mmol/L (IQR)” should be the entire line
Answer: The format of the Tables is not how we submitted them. We have adjusted the width of the columns of Table 1 to make sure the data are presented on 1 line.

Question 3: In all Tables; The description and explanation of the tables should appear below the table and not be part of the table title.
Answer: Thank you for noticing this, this format looks different from the format we submitted and we have changed this back now. The explanation of the tables is now below the tables again.

Question 4: Why do the authors decide to consider results with a p value of less than 0.1 as statistically significant (Usually, statistically significant values below 0.05 are taken)?
Answer: As explained in the method section (line 164 – line 171), we used a selection process to decide which sterols should be included in the final analyses. Since this was the first study to explore the associations between non-cholesterol sterol concentrations in breast milk and allergic outcomes, we did not want to exclude potentially interesting sterols with this selection process. Therefore, we
included all sterols that were significantly associated to an allergic outcome (p<0.05) or showed a trend towards significance (p<0.10). During the final logistic regression analyses (Section 3.3, Tables 5 and 6) we only considered results that were statistically significant (p<0.05). We have added the p-value we used for the final analysis in line 171 to clarify.

Question 5: Moreover, “p” should be in italics (i.e. p-value), please correct throughout the manuscript
Answer: Thank you for noticing this, we have corrected this throughout the manuscript.

Question 6: Line 304, 319; Please add references.
Answer: We have added references in these lines.